# Temperature Responsive Diblock Polymer Brushes as Nanoreactors for Silver Nanoparticles Catalysis

**DOI:** 10.3390/polym15081932

**Published:** 2023-04-19

**Authors:** Liang Yu, Ziwei Li, Chen Hua, Kaimin Chen, Xuhong Guo

**Affiliations:** 1State Key Laboratory of Chemical Engineering, School of Chemical Engineering, East China University of Science and Technology, Shanghai 200237, China; yuliang980213@foxmail.com; 2College of Chemistry and Chemical Engineering, Shanghai University of Engineering Science, Shanghai 201620, China; v2904602967@163.com; 3Wuxi Biologics, Wuxi 214013, China

**Keywords:** diblock polymer brush, temperature response, hydrogen bond, silver catalyst

## Abstract

Metal nanoparticles are widely used in catalysis. Loading metal nanoparticles into polymer brushes has aroused wide attention, but regulation of catalytic performance still needs to be improved. The novel diblock polymer brushes, polystyrene@sodium polystyrene sulfonate-b-poly (N-isopropylacrylamide) (PSV@PSS-b-PNIPA) and PSV@PNIPA-b-PSS with reversed block sequence, were prepared by surface initiated photoiniferter-mediated polymerization (SI-PIMP) and used as nanoreactors to load silver nanoparticles (AgNPs). The block sequence caused the difference of conformation and further affected the catalytic performance. PSV@PNIPA-b-PSS@Ag was found to be able to control the amount of AgNPs exposed to external reactant of 4-nitrophenol at different temperatures to achieve regulation of the reaction rate due to the hydrogen bonds and further physical crosslinking between PNIPA and PSS.

## 1. Introduction

Metal nanoparticles have attracted wide attention and can be applied in many fields such as catalysis, disease treatment, water purification, and imaging contrast agents [1,2,3,4]. However, metal nanoparticles easily agglomerate, which hinders the performance and practical applications. Loading metal nanoparticles into polymer systems [5,6] to suppress agglomeration, such as dendrimers [7], spherical polymer brushes [8,9,10,11,12] and microgels [13], has attracted many scientists’ interest due to their feasibility and broad prospects.

Among the various polymer systems, a spherical polymer brush (SPB) structure is considered to be the most useful tool to entrap a large amount of counterions (metal ions) in the brush layer based on the Donnan effect and electrostatic interactions, followed by the reduction of metal ions to obtain metal nanoparticles in situ [8,14]. Various spherical polyelectrolyte brushes, such as polystyrene@poly(2-aminoethyl methacrylate) (PS@PMPTAC) [14] and polystyrene@poly(acrylic acid) (PS@PAA) [15], were utilized to stabilize sliver nanoparticles (AgNPs), which demonstrated a good dispersion. AgNPs stabilized by PS@PMPTAC and PS@PAA showed high catalytic efficiency when used as the catalyst in reactions such as the reduction of 4-nitrophenol (4-NP). However, the catalytic reaction lacked control of the reaction rate at different temperatures because the temperature could not influence the distribution of mental nanoparticles in the above SPB. Lu [13,16] introduced temperature responsive crosslinked poly(N-isopropylacrylamide) (PNIPA) to successfully achieve regulation of catalytic performance of metal nanoparticles. It was proposed that the number of exposed metal nanoparticles on PNIPA could be changed by adjusting the temperature. However, the regulation was limited for a pure PNIPA-based system because of its constant LCST. The LCST of PNIPA could be altered simply by introduction of hydrophilic monomers [17]. Cang [18] selected AA as the hydrophilic comonomer, and PS@P(AA-co-NIPA) was prepared. It was found that low content of PAA could remarkably adjust the LCST, but the LCST was out of control at high content of PAA when AA was copolymerized in the structure. To fine tune the LCST, diblock copolymer is selected as an alternative structure that can be realized by living/controlled free radical polymerization, such as atom transfer radical polymerization (ATRP) [19], reversible addition-fragmentation chain transfer (RAFT) [20], nitroxide-mediated polymerization (NMP) [21], and photoiniferter-mediated polymerization (PIMP) [22]. Among them, PIMP has attracted more and more attention due to its mild reaction conditions: it can react at room temperature without intense reaction conditions. 4-vinylbenzyl N,N-diethyldithiocarbamate (VBDC) was successfully prepared for the radical copolymerization of VBDC with styrene by Otsu [23] in 1986. It was then widely used as an iniferter (initiator-transfer agent-terminator) of PIMP. Cao [24] used this method to prepare dual responsive diblock polymer brushes, PSV@PAA-b-PNIPA, and PSV@PNIPA-b-PAA with comparable PAA and PNIPA components. Both temperature and pH responses were maintained, and various conformations could be obtained by adjusting the block sequence. Wang [25] used N-hydroxyethyl acrylamide (HEAA), an excellent anti protein adsorption material, and AA to prepare PSV@PHEAA-b-PAA and PSV@PAA-b-PHEAA. By utilizing the different characteristics of two polymers in protein adsorption and anti-fouling, controllable loading of proteins was achieved. From the above, it can be seen that using PIMP to prepare diblock polymer brushes could leverage the characteristics of each polymer and expand the functionality of the new system. Meanwhile, it is worth noting that the use of PIMP to prepare spherical diblock polymer brushes for metal loading is still rare.

In this study, sodium styrenesulfonate (SS), a fully ionized monomer contributing to the enrichment of counterions [26], and NIPA were selected to prepare diblock polymer brushes with reversed block sequence: PSV@PSS-b-PNIPA and PSV@PNIPA-b-PSS. The temperature responsive behavior was studied in detail. Then, diblock polymer brushes were used as nanoreactors to immobilize AgNPs in situ. The catalytic ability and characteristics of diblock polymer brushes loading AgNPs with different structures at different temperatures were studied by selecting the reduction of 4-NP as a model catalysis reaction.

## 2. Materials and Methods

### 2.1. Materials

Sodium 4-vinylbenzenesulfonate (SS; Adamas-beta) (Shanghai, China), *N*,*N*-methylenebis (acrylamide) (BIS; Adamas-beta) (Shanghai, China), sodium chloride (NaCl; Sinopharm) (Shanghai, China), sodium dodecyl sulfate (SDS; J&K) (Shanghai, China), potassium persulfate (KPS; J&K) (Shanghai, China), sodium diethyldithiocarbamate (DDTC; J&K) (Shanghai, China), 4-vinybenzyl chloride (CMS; J&K) (Shanghai, China), sodium borohydride (NaBH_4_; Sinopharm) (Shanghai, China), 4-nitrophenol (4-NP; Meryer) (Shanghai, China), silver nitrate (AgNO_3_; ACROS) (Tokyo, Japan), 2-hydroxy-40-hydroxyethoxy-2-methyl propiophenone (HMP; Ciba) (Basel, Switzerland), and methacryloyl chloride (MC; Adamas-beta) (Shanghai, China) were used without further purification. Styrene (St; Sinopharm) (Shanghai, China) was distilled under reduced pressure to remove the inhibitor. N-isopropyl acrylamide (NIPA; Aladdin) (Shanghai, China) was recrystallized from hexane and stored in a dark environment at 4 °C. N, N-diethyldithiocarbamate (VBDC) was synthesized according to previous publications, and the specific steps will be explained in detail below [23].

### 2.2. Synthesis of VBDC

The synthesis of VBDC was achieved through the substitution reaction of CMS and DDTC. The entire reaction process in VBDC synthesis needed to avoid light. Firstly, CMS (2.54 g) was dissolved in anhydrous ethanol (9 mL) and transferred to a 100 mL three-necked flask with magnetic stirring. DDTC (5 g) was dissolved in anhydrous ethanol (25 mL). Then, it was added to a constant pressure drip funnel. The entire device was pumped and filled with nitrogen gas three times to ensure that subsequent reactions took place in inert gas. The three necked flask was placed in an ice water bath, and then, a constant pressure drip funnel was opened to control the addition of CMS ethanol solution to the three necked flask at a rate of 6 s per drop. After the liquid was added completely, the reaction continued for 30 min. Then, the ice water bath was removed. The reaction continued at room temperature for 20 h before ending the reaction. The product was collected for subsequent purification. Purification operations included extraction, water washing, drying, rotary steaming, recrystallization, and vacuum drying. Finally, pure VBDC was obtained and stored in low temperature and under dark conditions.

### 2.3. Synthesis of PS and PSV Cores

Figure 1 shows the synthesis of PS and PSV cores. PS cores were obtained by conventional emulsion polymerization of styrene. Styrene (4 g), SDS (0.08 g), KPS (0.24 g), and 96 mL of H_2_O were added into 250 mL three-necked flask with a stirring speed of 300 rpm. After pumping and filling nitrogen gas three times, the oil bath temperature was set to 80 °C, and it continued heating for 1 h after reaching 80 °C. The obtained emulsion was dialyzed with ultrapure water until the conductivity remained unchanged. Then, PS cores were obtained. PSV cores were obtained by seeded soap-free emulsion copolymerization, and the entire experimental process was conducted in the dark [27]. PS cores (0.25 g), KPS (0.025 g), and 93 mL of H_2_O were added into 250 mL three-necked flask with a stirring speed of 300 rpm. VBDC (0.0125 g) was dissolved with styrene (0.4 g) and then transferred to a constant pressure drip funnel. When the oil bath temperature reached 70 °C, the constant pressure drip funnel was controlled to drip liquid into a three-necked flask at a rate of 6 s per drop under a nitrogen atmosphere. After the liquid was added completely, the system continued to react at 70 °C for 5 h to obtain PSV cores. The cores were also dialyzed with ultrapure water until the conductivity remained unchanged. The obtained PSV cores needed to be stored away from light. The solid content of PS and PSV was measured using a high-temperature oven at 120 °C, and each sample needed to be tested three times.

### 2.4. Synthesis of Diblock Polymer Brushes

Figure 1 also shows the synthesis of diblock polymer brushes PSV@PSS-b-PNIPA and PSV@PNIPA-b-PSS. PSV cores (0.5 g), SS monomer (1.0 g), and water (20 g) were added into a photo-reactor with gentle magnetic stirring. The mixture was UV (200–600 nm, 175 W) irradiated for 3.5 h under a nitrogen atmosphere to obtain PSV@PSS. Finally, PSV@PSS (0.3 g), NIPA monomer (0.6 g), and water (10 g) were mixed and UV irradiated for 3 h under a nitrogen atmosphere to obtain PSV@PSS-b-PNIPA. PSV@PNIPA-b-PSS was obtained by a similar method by the reverse order of SS and NIPA addition, just as shown in Figure 1. Firstly, PSV cores (0.5 g), NIPA monomer (1.0 g), and water (20 g) were added into a photo-reactor with gentle magnetic stirring and UV irradiated for 3 h under nitrogen atmosphere to obtain PSV@PNIPA. Finally, PSV@PNIPA (0.3 g), SS monomer (0.6 g), and water (10 g) were mixed and UV irradiated for 3.5 h under a nitrogen atmosphere to obtain PSV@PNIPA-b-PSS. After the preparation of the polymer brushes mentioned above, dialysis with ultrapure water was required until the conductivity remains unchanged. All samples needed to be stored away from light. At the same time, their solid content was measured using a high-temperature oven at 120 °C, and each sample needed to be tested three times. The solid content of polymer brushes was also measured.

### 2.5. Immobilization of AgNPs on Diblock Polymer Brushes

Figure 1 also showed the synthesis of PSV@PSS-b-PNIPA@Ag. A solution of 5 mL AgNO_3_ (16 mM) was added dropwise to PSV@PSS-b-PNIPA (0.1 g) to replace Na^+^ with Ag^+^. Subsequently, the adsorbed Ag^+^ on the shell was reduced by 5 mL NaBH_4_ solution (160 mM), and then, free AgNPs were removed by ultrafiltration to obtain PSV@PSS-b-PNIPA@Ag. PSV@PNIPA-b-PSS@Ag was prepared via a similar route as shown in Figure 1. A solution of 5 mL AgNO_3_ (16 mM) was added dropwise to PSV@PNIPA-b-PSS (0.1 g) to replace Na^+^ with Ag^+^. Subsequently, the adsorbed Ag^+^ on the shell was reduced by 5 mL NaBH_4_ solution (160 mM) Then, free AgNPs were removed by ultrafiltration to obtain PSV@PNIPA-b-PSS@Ag.

### 2.6. Reduction of 4-NP to 4-AP

Firstly, 1.5 mL of 4-NP (0.2 mM) and 1.5 mL of NaBH_4_ (20 mM) were added to a glass vessel. Then, a given amount of AgNPs was added to it. Immediately, UV spectra of the sample were taken in the range of 200–500 nm for 10 min. The rate constant was determined by recording the variation of adsorption peak at 400 nm with time.

### 2.7. Characterization

The diameter of diblock polymer brushes was determined by dynamic light scattering (DLS, NICOMP 380, Entegris, Billerica, MA, USA) with the measurement angle of 90° and collected by the autocorrelator at the preset temperature. When using DLS to measure the size of polymer brushes at different temperatures, the initial temperature was set at 15 °C, with intervals of 3 °C, until 50 °C was reached. In order to obtain stable data at different temperatures, a temperature balance of 8 min was required before each measurement. The particle size at any temperature was measured three times. Transmission electron microscope (TEM) images were taken on a JEM-1400 (JEOL, Tokyo, Japan). UV-Vis absorption spectra of samples from 200 nm to 500 nm were recorded on a UV-1900i (Shimadzu, Tokyo, Japan). The FT-IR spectra were recorded between 4000 cm^−1^ and 400 cm^−1^ by a potassium bromide (KBr) tableting method on a Fourier transform-infrared spectrometer (Negoli, Waltham, MA, USA). The ^1^H NMR spectra were recorded on a 400 MHz Super Conducting Fourier NMR Spectrometer (Brooke Technology Co., Ltd., Basel, Switzerland). Ag content was measured by inductively coupled plasma optical emission spectrometer (ICP-OES, Agilent 725, VARIAN, Palo Alto, CA, USA). Elemental analysis was obtained by an elemental analyzer (VARIO EL CUBE, Elementar Analysensysteme GmbH, Langenselbold, Germany).

## 3. Results and Discussion

### 3.1. Synthesis of Diblock Polymer Brushes

The ^1^H NMR spectra (Appendix A) of VBDC were consistent with the literature [23], confirming the successful synthesis of VBDC. FT-IR was applied to confirm the existence of functional groups (Figure 2a). Compared to PS, peaks at 1209 cm^−1^, 1268 cm^−1^, and 1415 cm^−1^ were attributed to the adsorption of C=S, C-S, and C-N in VBDC. The peaks at 1181 cm^−1^ and 1039 cm^−1^ were observed for PSV@PSS, PSV@PSS-b-PNIPA, and PSV@PNIPA-b-PSS, which could be ascribed to the stretching vibration of S=O and S-O in PSS. Peaks at 1544 cm^−1^ and 1653 cm^−1^ corresponding to the amide I band (due to the C=O stretching vibration) and the amide II band (a combination of the C-N bending vibration and C-N-H stretching vibration) of PNIPA for PSV@PSS-b-PNIPA, PSV@PNIPA, and PSV@PNIPA-b-PSS proved the existence of PNIPA.

The morphology and diameter of diblock polymer brushes were characterized by TEM and DLS. PSV cores (134 nm) were larger than PS (91 nm) because of copolymerization of VBDC and styrene (Figure 2b,c). The size of PSV@PSS and PSV@PSS-b-PNIPA was 268 nm and 510 nm, respectively (Figure 2c); the size of PSV@PNIPA and PSV@PNIPA-b-PSS was 241 nm and 264 nm (Figure 2d), respectively. TEM images showed a similar size for PS core and PSV. After grafting of polymers, the TEM images showed core-shell structure with a dark black core and a blurry irregular shell as depicted in Figure 2b. The observed size for TEM was smaller than that of DLS, which was caused by the collapse of the polymer chains [28].

The elemental analysis of diblock polymer brushes was accomplished to quantitatively obtain the C, N, and S element contents, as shown in Table 1. For PSV@PSS-b-PNIPA, the molar ratio of PNIPA:PSS was 9.9:1, bigger than the diameter ratio (1.8:1) of PNIPA:PSS because PSS was fully ionized and more hydrophilic than PNIPA. For PSV@PNIPA-b-PSS, the molar ratio of PSS:PNIPA was 2.22:1, which was an interesting phenomenon because DLS indicated that the diameter ratio of PSS:PNIPA was 0.21:1, which was much smaller. Cao [29] reported that the formation of hydrogen bonds between PNIPA and PSS could increase self-healing and adhesion abilities of the hydrogel. The abnormal diameter of PSV@PNIPA-b-PSS may be caused by the formation of hydrogen bond between PSS and PNIPA, which will be verified later.

### 3.2. Ionic Strength Responsive Properties of Diblock Polymer Brushes

External ions (NaBH_4_) need to be introduced in subsequent catalytic experiments, so the stability of polymer brushes in different ionic strength was studied. The ionic strength was adjusted by adding suitable concentration of sodium chloride (NaCl).

For PSV@PSS, the diameter decreased from 245 nm to 170 nm when ionic strength increased from 0.01 mM to 100 mM (Figure 3a). The increase of ionic strength led to the increase of counter ion concentration between PSS chains, which decreased electrostatic repulsion force between chains, leading to a decrease in particle size. For PSV@PSS-b-PNIPA, the diameter decreased from 450 nm to 295 nm when ionic strength increased from 0.01 mM to 100 mM (Figure 3a). It was worth noting that the decrease of particle size mainly occurred when the ionic strength increased from 0.01 mM to 1mM. The PNIPA block would not affect the ionic strength response of PSS for PSV@PSS-b-PNIPA. The diameter of PSV@PNIPA remained almost unchanged with ionic strength because PNIPA was neutral and would not be subject to the electrostatic shielding effect (Figure 3b). For PSV@PNIPA-b-PSS, the diameter decreased from 245 nm to 193 nm when ionic strength increased from 0.01 mM to 100 mM because of the existence of PSS. It was worth noting that the PDI of all polymer brushes under different ionic strength was less than 0.05, which indicates that the polymer brushes maintained good dispersion and stability under different ionic strength.

### 3.3. Temperature Responsive Properties of Diblock Polymer Brushes

The diameter of diblock polymer brushes was temperature-dependent due to the existence of PNIPA. Therefore, the change of diameter of different polymer brushes with temperature was studied. PSV@PNIPA showed typical LCST at ~30 °C (Appendix A). For diblock polymer brushes, PSV@PSS-b-PNIPA showed a temperature-dependent size change (Figure 4a), and the corresponding LCST was about 37 °C, which was higher than that of PSV@PNIPA. The hydrophilic PSS made PSV@PSS-b-PNIPA more hydrophilic than PSV@PNIPA, which then raised the LCST a little. For PSV@PNIPA-b-PSS, it showed a LCST at 34 °C (Figure 4b). It was worth noting that the diameter of PSV@PNIPA-b-PSS only decreased ~40 nm, which was much smaller than that of PSV@PNIPA, with ~70 nm.

For above three temperature responsive polymer brushes, the amide bond of PNIPA could be used as a hydrogen acceptor and hydrogen donor at the same time, so the hydrogen bond could also be formed inside PNIPA. The heating process promoted the chain contraction and made the shell quickly undergo volume phase transition. However, during the cooling process, the intramolecular hydrogen bond caused shell swelling delay, forming a lag ring during the heating–cooling process [30,31].

To determine the reasons for large differences in diameter shrinkage between PSV@PNIPA and PSV@PNIPA-b-PSS, the temperature response of diameter of PSV@PNIPA-b-PSS at different salt concentration was determined, as shown in Figure 5a. Without additional salt (0 mM), it showed normal LCST behaviors. At a salt concentration of 100 mM, when the temperature increased, the diameter of PSV@PNIPA-b-PSS decreased first and increased at temperatures higher than 33 °C. The size increase stopped at 310 nm at 50 °C with a narrow PDI, indicating that the size increase was not caused by aggregation, but probably brought by the conformation change of the diblock polymer shell. Combined with the above inconsistencies between the size and the elemental analysis results, it was proposed that hydrogen bonds exist between PSS and PNIPA. During the synthesis process of PSV@PNIPA-b-PSS, PSS grew from the end of PNIPA. There were many sites in PNIPA to form hydrogen bonds with PSS, resulting in entanglement of PSS chain along the PNIPA chain. It was worth noting that it was easy to form physical entanglement to form a complex network structure when PSS grew inside PNIPA. This could also explain the results of element analysis. Although the molar ratio of PSS:PNIPA was 2.22:1, the length of PSS chain only increased to 318 nm after the hydrogen bond was destroyed (Figure 5a). Only part of the PSS chain departed from the PNIPA network, and other PSS were still trapped in the network structure. The proposed conformation of PSV@PNIPA-b-PSS under different conditions is shown in Figure 5b.

The proposed conformations could explain the lower shrinkage of PSV@PNIPA-b-PSS than PSV@PNIPA. When the diameter of PSV@PNIPA-b-PSS decreased with temperature, the distance between PSS chains became shorter and the electrostatic repulsion force increased, which limited the further shrinkage of network structure.

### 3.4. Immobilization of Ag on Diblock Polymer Brushes

Diblock polymer brushes, PSV@PSS-b-PNIPA, and PSV@PNIPA-b-PSS were used to prepare PSV@PSS-b-PNIPA@Ag and PSV@PNIPA-b-PSS@Ag.

The diameter and PDI of diblock polymer brushes loading AgNPs are shown in Table 2 (ΔD represents the size difference of diblock polymer brushes before and after loading of AgNPs). After immobilization of Ag, the diameter of diblock polymer brushes decreased. The structure and morphology of PSV@PSS-b-PNIPA@Ag and PSV@PNIPA-b-PSS@Ag were characterized by TEM (Figure 6). They still maintained a relatively spherical structure with uniform size. Some black dots could be observed around the particles, which were AgNPs generated in situ by NaBH_4_ reduction for Ag^+^. AgNPs were uniformly distributed and smaller in size due to the fact that their nucleation occurred at multiple sites on the particle surface, resulting in smaller particles than those grown in solution under the same conditions.

The structure of polymer has a strong impact on the amount, size, and shape of the synthesized nanoparticles, which could be also found from TEM images (Figure 6). PSV@PNIPA-b-PSS@Ag had a network structure, and the AgNPs were uniformly distributed in the outer region of the PSV cores (Figure 6b). However, PSV@PSS-b-PNIPA@Ag had a normal diblock brush structure, and AgNPs were distributed on the PSV cores (Figure 6a). The size of AgNPs was smaller and the distribution of AgNPs was narrower for PSV@PNIPA-b-PSS@Ag compared to PSV@PSS-b-PNIA@Ag. The results of ICP showed that the content of Ag element of PSV@PSS-b-PNIPA@Ag and PSV@PNIPA-b-PSS@Ag was 35 mg/L and 33 mg/L, respectively. The above results indicate that the diblock polymer brushes could be used as an effective nanoreactor for the preparation of AgNPs with uniform particle size, and the structure of 9eblock polymer brushes had a strong impact on the amount, size, and shape of AgNPs.

The change of diameter of 9eblock polymer brushes after loading of AgNPS with temperature was studied, as shown in Figure 7. PSV@PSS-b-PNIPA@Ag showed temperature-dependent size change (Figure 7a), and the corresponding LCST was about 36 °C, which was slightly lower than that (37 °C) of PSV@PSS-b-PNIPA. PSV@PNIPA-b-PSS showed a LCST of 30 °C, which was lower than that (34 °C) of PSV@PNIPA-b-PSS (Figure 7b). After the loading of AgNPs, the two diblock polymer brushes still maintained good temperature response with a lower LCST.

### 3.5. Catalytic Performance of AgNPs

The catalytic activity of AgNPs was investigated by reduction of 4-nitrophenol (4-NP) to 4-aminophenol (4-AP) in NaBH_4_. Appendix A shows the UV spectra for the reduction of 4-NP using diblock polymer brushes loading AgNPs. The specific amounts of substances in the reduction experiment are detailed in Appendix A.

Figure 8 displays the reduction of 4-NP of PSV@PSS-b-PNIPA@Ag and PSV@PNIPA-b-PSS@Ag at different temperatures. For PSV@PSS-b-PNIPA@Ag, the apparent kinetic constant (k_app_) increased with temperature because AgNPs were mainly loaded on the first PSS block, and the structure of the second PNIPA block at different temperature did not influence the distribution of AgNPs (Figure 8a,b). The reduction of 4-NP by PSV@PNIPA-b-PSS@Ag was also investigated (Figure 8c,d). An interesting phenomenon was noticed. The k_app_ normally increased when temperature changed from 293 K to 303 K. However, the k_app_ decreased a lot when the temperature changed from 303 K to 313 K. When PNIPA shrunk, the whole network structure would shrink, so the number of AgNPs exposed to the reactants would decrease sharply, which made k_app_ decrease significantly at 313 K. When the temperature was over 313 K, the network of PSV@PNIPA-b-PSS@Ag had completely shrunk, so the structure of PSV@PNIPA-b-PSS@Ag remained unchanged. Therefore, k_app_ increased when temperature changed from 313 K to 323 K due to the endothermic reaction.

## 4. Conclusions

Different temperature responsive diblock polymer brushes were successfully synthesized and used as the nanoreactor to load AgNPs. The diameter ratio and molar ratio of PNIPA/PSS for PSV@PSS-b-PNIPA was 1.9:1 and 9.9:1, respectively. PSS and PNIPA for PSV@PNIPA-b-PSS formed hydrogen bonds, resulting in remarkable physical crosslinking, which decreased the diameter ratio of PSS/PNIPA for PSV@PNIPA-b-PSS to 0.21:1 when the molar ratio was 2.22:1. The physical crosslinked PSV@PNIPA-b-PSS could control the amount of AgNPs exposed to external reactants at different temperatures to achieve regulation of the reaction rate. This work provides a method of loading metal nanoparticles with regulated catalytic performance.

## Figures and Tables

**Figure 1 polymers-15-01932-f001:**
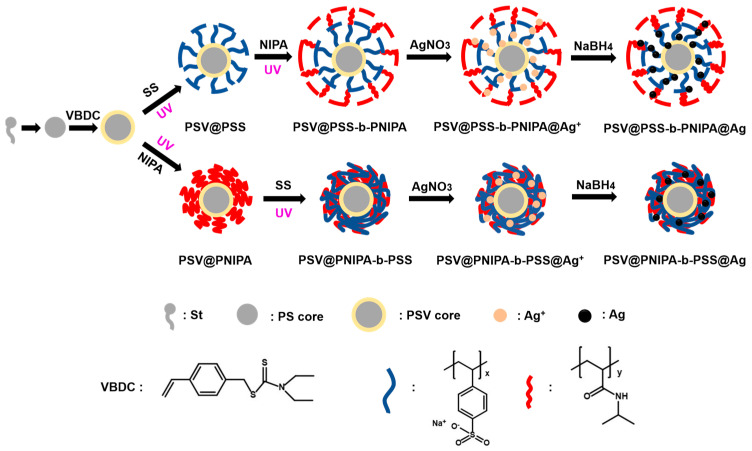
Schematic illustration of PSV@PSS-b-PNIPA@Ag and PSV@PNIPA-b-PSS@Ag.

**Figure 2 polymers-15-01932-f002:**
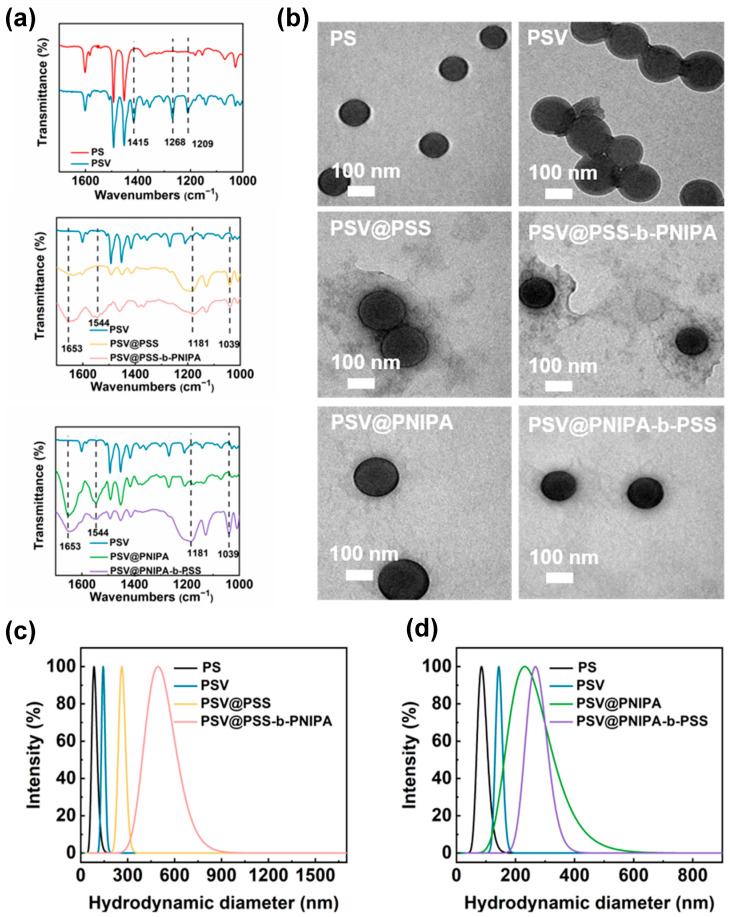
(**a**) FT-IR spectra and (**b**) TEM images and (**c**,**d**) size distribution of PS, PSV, PSV@PSS, PSV@PNIPA, PSV@PSS-b-PNIPA and PSV@PNIPA-b-PSS.

**Figure 3 polymers-15-01932-f003:**
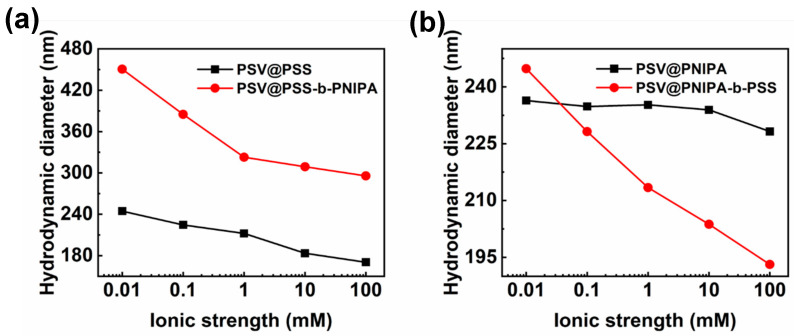
Particle size of (**a**) PSV@PSS, PSV@PSS-b-PNIPA, (**b**) PSV@PNIPA and PSV@PNIPA-b-PSS under different ionic strengths.

**Figure 4 polymers-15-01932-f004:**
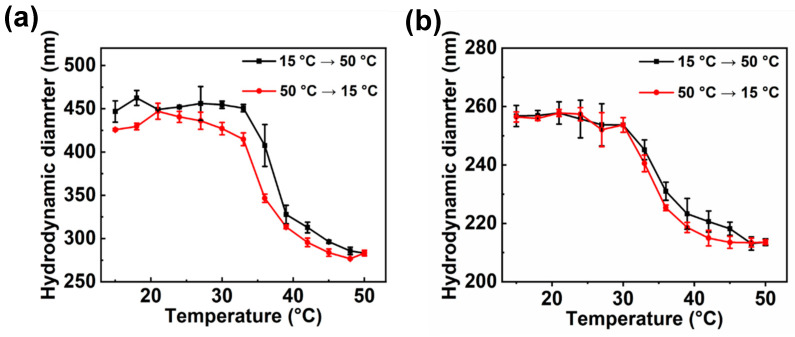
Diameter of (**a**) PSV@PSS-b-PNIPA and (**b**) PSV@PNIPA-b-PSS with temperature.

**Figure 5 polymers-15-01932-f005:**
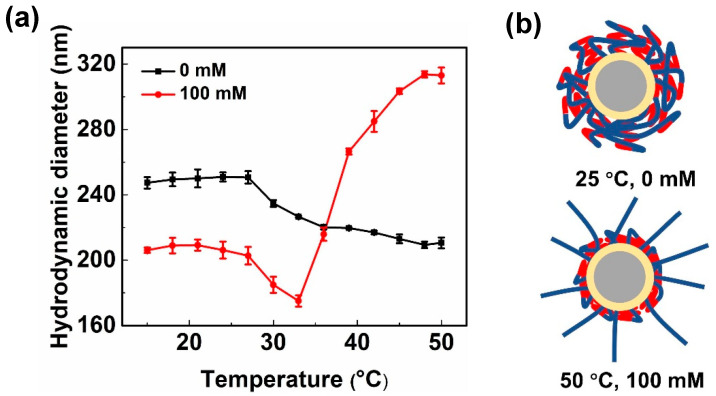
(**a**) Diameter with temperature at different ionic strength and (**b**) schematic illustration of PSV@PNIPA-b-PSS.

**Figure 6 polymers-15-01932-f006:**
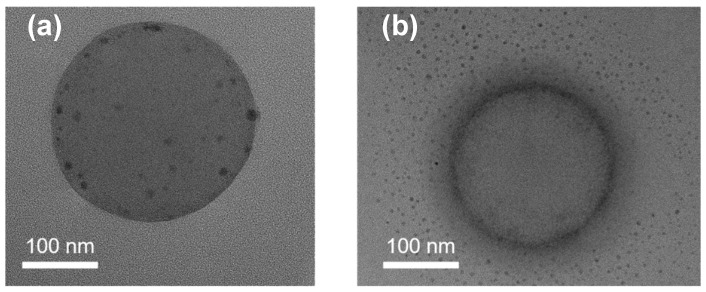
TEM images of (**a**) PSV@PSS-b-PNIPA@Ag and (**b**) PSV@PNIPA-b-PSS@Ag.

**Figure 7 polymers-15-01932-f007:**
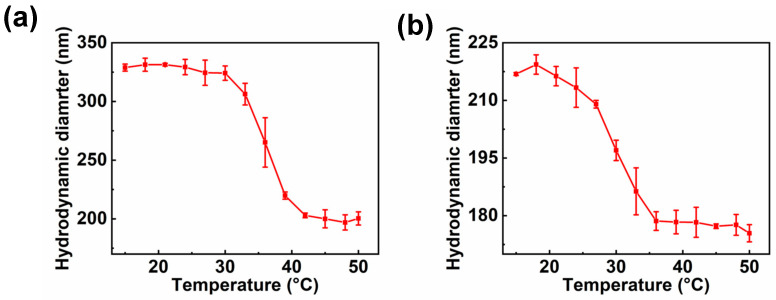
Diameter of (**a**) PSV@PSS-b-PNIPA@Ag and (**b**) PSV@PNIPA-b-PSS@Ag with temperature.

**Figure 8 polymers-15-01932-f008:**
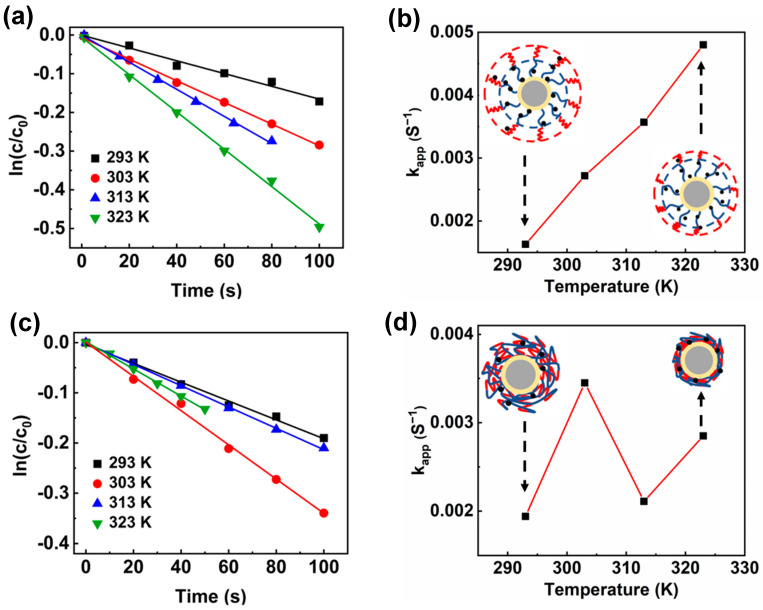
Kinetics of the reduction of 4-NP and relationship between apparent kinetic constant (k_app_) and temperature (K) of (**a**,**b**) PSV@PSS-b-PNIPA@Ag and (**c**,**d**) PSV@PNIPA-b-PSS@Ag.

**Table 1 polymers-15-01932-t001:** Elemental analysis of diblock polymer brushes.

	C%	N%	H%	S%
PS	92.36	-	7.64	-
PSV	86.16	1	8.69	4.15
PSV@PSS	60.50	0.52	6.318	8.215
PSV@PSS-b-PNIPA	57.32	7.19	8.223	2.886
PSV@PNIPA	77.08	3.43	7.760	3.417
PSV@PNIPA-b-PSS	56.12	1.63	6.086	8.075

**Table 2 polymers-15-01932-t002:** Diameter and PDI of 9eblock polymer brushes loading AgNPs.

	Diameter (nm)	ΔD (nm)	PDI
PSV@PSS-b-PNIPA@Ag	394	−116	0.10
PSV@PNIPA-b-PSS@Ag	235	−29	0.12

## Data Availability

Data presented in this study are available on request from the corresponding author.

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
