# Peer review of "Temperature Responsive Diblock Polymer Brushes as Nanoreactors for Silver Nanoparticles Catalysis"

_polymers, 2023, doi:10.3390/polym15081932_

Round 1

Reviewer 2 Report

The paper presents novel and interesting information and can be accepted for publication after some important issues will be clarified.

Please add appropriate information about the synthesis of PSV cores, what does this abbreviature mean? Which type of SI-PIMP was used in this case and why? Please add information on the condition of the grafting polymerization, which wavenumber of the UV-irradiation was used. Why SI-PIMP was applied in this case, "due to its mild reaction conditions" is not clear to me.

It is well-known that polymer matrix has a strong impact on the amount, size, and shape of the synthesized nanoparticles. This information was omitted in the paper. Please add appropriate discussion. Ideally, TGA should be applied to compare the amounts of silver nanoparticles in both of the samples.

What about the impact of the silver nanoparticles on the LCST? As rule, grafted polymer brushes with silver nanoparticles have lower LCST than brushes without silver nanoparticles. 

Finally please cite highly relevant papers where similar systems were developed:

https://doi.org/10.1016/j.colsurfa.2022.128525

https://doi.org/10.1016/j.apsusc.2018.09.033

Round 2

Reviewer 2 Report

The paper is perfect after revision and can be accepted in its present form.